# Existence and uniqueness of neutral functional differential equations with sequential fractional operators

**Rabah Debbar[1], Hamid Boulares[2], Abdelkader Moumen [3]\*, Tariq Alraqad[3], Hicham Saber[3]**

**1** University of 8 May 1945 Guelma, Guelma, Algeria, **2** Laboratory of Analysis and Control of Differential Equations "ACED", Department of Mathematics, Faculty MISM, University of 8 May 1945 Guelma, Guelma, Algeria, **3** Department of Mathematics, College of Science, University of Ha'il, Ha'il, Saudi Arabia

\* mo.abdelkader@uoh.edu.sa

**Data Availability Statement:** All relevant methods and materials are contained within the manuscript.

**Funding:** The author(s) received no specific funding for this work. This research has been

## Abstract

In this research paper, we investigate the existence and uniqueness of solutions for neutral functional differential equations with sequential fractional orders, specifically involving the $\mathbb{G}$-Caputo operator. To obtain the desired results, we employ the Banach fixed point theorem (BFPT), a nonlinear variation of the Leray-Schauder fixed point theorem (SFPT), and the Krasnoselski fixed point theorem (KFPT). Additionally, we provide illustrative examples that demonstrate the key findings. Furthermore, we address a scenario where an initial value integral condition is considered.

## 1 Introduction

The objective of the axes of this study first is a direct generalization of the research [1] which refers to the investigation of the EUS to the initial value problem (IVP) involving neutral functional differential equations (FDEs) with sequential fractional orders and the $\mathbb{G}$-Caputo operator

$$\mathcal{D}_{\mathfrak{a}}^{\alpha_1;\mathbb{G}}[\mathcal{D}_{\mathfrak{a}}^{\alpha_2;\mathbb{G}}\mathfrak{u}(r) - \hbar(r, \varkappa_r)] = \mathcal{F}(r, \mathfrak{u}_r), \quad r \in J \coloneqq [\mathfrak{a}, \mathfrak{b}], \tag{1}$$

$$\mathfrak{u}(r) = \phi(r), \quad r \in [\mathfrak{a} - \tau, \mathfrak{a}], \tag{2}$$

$$\mathcal{D}_{\mathfrak{a}}^{\alpha_2;\mathbb{G}}\mathfrak{u}(\mathfrak{a}) = \eta \in \mathbb{R}, \tag{3}$$

where $\mathcal{D}_{\mathfrak{a}}^{\alpha_1;\mathbb{G}}, \mathcal{D}_{\mathfrak{a}}^{\alpha_2;\mathbb{G}}$ denote $\mathbb{G}$-Caputo fractional derivatives, $0 < \alpha_1, \alpha_2 < 1, \mathcal{F}, \hbar : J \times C([-\tau, 0], \mathbb{R}) \to \mathbb{R}$ are functions, and $\phi \in C([\mathfrak{a} - \tau, \mathfrak{a}], \mathbb{R})$.

Let $\mathfrak{u}$ be defined on $[\mathfrak{a} - \tau, \mathfrak{b}]$: Then for $r \in J$, we denote by

$$\mathfrak{u}_r \in C_\tau \coloneqq C([-\tau, 0], \mathbb{R}) \text{ given by } \mathfrak{u}_r(\theta) = \mathfrak{u}(r + \theta), \quad \theta \in [-\tau, 0].$$

Differential functional equations play a crucial role across various domains, including control theory, neural networks, and epidemiology [2]. Specifically, delay differential equations prove

funded by Scientific Research Deanship at University of Ha'il - Saudi Arabia through project number RG-23 036.

invaluable in understanding the dynamics of real populations, handling both finite and infinite delay terms. The presence of delays in derivatives poses challenges, prompting an alternative approach through the exploration of neutral functional differential equations. Additionally, fractional derivatives become instrumental in capturing hereditary as well as memory effects in diverse materials and processes. Thus, making the investigation of functional neutral differential equations that involve fractional derivatives a vital area of research. Hybrid problems, which involve the combination of different mathematical techniques or models to solve complex physical phenomena, have applications in various fields. Some examples of physical applications of hybrid problems include Fluid-structure interaction, electromagnetic-structural interaction, thermo-mechanical analysis, coupled heat and mass transfer, multiphase flow with chemical reactions, biomechanics. Comprehensive information on this topic can be found in referenced texts [3–7].

Significant progress in fractional calculus, especially concerning initial value problems (IVPs) of fractional differential equations (FDEs), has been made in recent years. Contributions from various authors, such as Kilbas *et al.* [8], Lakshmikantham *et al.* [9], Miller and Ross [10], Podlubny [11], Samko *et al.* [12], Diethelm [13], along with related papers [14–23], have enriched this field. Riemann-Liouville and Caputo type fractional derivatives dominate this research, while the Hadamard derivative introduces a unique perspective with its logarithmic function containing an arbitrary exponent. See [24–26] and references therein for a comprehensive descriptions of the Hadamard fractional integral and derivative.

In a previous study [27–30], researchers focused on an IVP involving fractional functional and neutral functional differential equations with infinite delay of the Riemann-Liouville type. Subsequent investigations [31] extended to IVPs concerning functional and neutral functional differential equations, including fractional order of Hadamard type. Another study [32] addressed an initial value problem involving retarded functional Caputo-type fractional impulsive differential equations with variable moments.

Fixed-point theory, a mathematical branch, explores the existence and properties of fixed points in mappings or functions. With applications across mathematics and various fields like economics, computer science, physics, engineering, and social sciences, fixed-point theory provides a framework for studying solutions to equations and mappings. The current research focuses on a novel class of sequential fractional neutral functional differential equations, particularly of the $\mathbb{G}$-Caputo type. Utilizing fixed point theorems by Banach and Krasnoselskii [33] together with the nonlinear version of Leray-Schauder type introduced by [34], this study advances our understanding of these equations.

The paper's remaining sections are organized as follows: Section 2 reviews crucial preliminaries for subsequent analysis. Section 3 examines the existence and uniqueness of solutions for the presented problem in Eqs (1) and (2). Section 4 presents the existence results for the problem. Section 5 introduces a generalization incorporating an initial value integral condition, along with illustrative examples showcasing the study's findings. Finally, Section 6 constructs additional illustrative examples to reinforce the obtained results.

## 2 Essentiel preliminaries

In the following section, we will introduce the notation, definitions, and preliminary facts that are essential for the subsequent analysis. We denote by $C(J, \mathbb{R})$ the Banach space consisting of all continuous functions from the interval $J$ to the real numbers $\mathbb{R}$. The norm associated with this space is given by

$$\|\mathfrak{u}\|_\infty := \sup\{|\mathfrak{u}(r)| : r \in J\}.$$

Also $C_\tau$ is endowed with norm

$$\|\phi\|_C := \sup\{|\phi(\theta)| : -\tau \le \theta \le 0\}.$$

Let $\mathbb{G} : [\mathfrak{a}, \mathfrak{I}] \to \mathbb{R}$ be increasing and $\mathbb{G}'(r) \ne 0, \forall r$.

**Definition 1.1** ([8, 35]). *The $\mathbb{G}$-Riemann-Liouville fractional integral ($\mathbb{G}$-RLFI) of order $\alpha > 0$ for a CF $\mathfrak{u} : [\mathfrak{a}, \mathfrak{I}] \to R$ is referred to as*

$$\mathcal{I}_\mathfrak{a}^{\alpha;\mathbb{G}}\mathfrak{u}(r) = \frac{1}{\Gamma(\alpha)} \int_\mathfrak{a}^r (\mathbb{G}(r) - \mathbb{G}(s))^{\alpha-1} \mathbb{G}'(s)\mathfrak{u}(s)ds.$$

**Definition 1.2** ([8, 35]). *The $\mathbb{G}$-Caputo fractional derivative ($\mathbb{G}$-CFD) of order $\alpha > 0$ for a CF $\mathfrak{u} : [\mathfrak{a}, \mathfrak{I}] \to R$ is the aim of*

$$\mathcal{D}_\mathfrak{a}^{\alpha;\mathbb{G}}\mathfrak{u}(r) = \frac{1}{\Gamma(n - \alpha)} \int_\mathfrak{a}^r (\mathbb{G}(r) - \mathbb{G}(s))^{n-\alpha-1} \mathbb{G}'(s)\partial_\mathbb{G}^n\mathfrak{u}(s)ds, \ \ r > \mathfrak{a}, \ \ \alpha \in (n-1, n),$$

*where* $\partial_\mathbb{G}^n = \left(\frac{1}{\mathbb{G}'(r)}\frac{d}{dr}\right)^n, \ \ n \in \aleph$

**Lemma 1.3** ([8, 35]). *Let $q, \ell > 0$, and $\mathfrak{u} \in \mathcal{C}([\mathfrak{a}, \mathfrak{b}], \mathbb{R})$. Then, $\forall r \in [\mathfrak{a}, \mathfrak{b}]$, and by assuming $F_\mathfrak{a}(r) = \mathbb{G}(r) - \mathbb{G}(\mathfrak{a})$, we have*

1. $\mathcal{I}_\mathfrak{a}^{q;\mathbb{G}}\mathcal{I}_\mathfrak{a}^{\ell;\mathbb{G}}\mathfrak{u}(r) = \mathcal{I}_\mathfrak{a}^{q+\ell;\mathbb{G}}\mathfrak{u}(r)$,

2. $\mathcal{D}_\mathfrak{a}^{q;\mathbb{G}}\mathcal{I}_\mathfrak{a}^{q;\mathbb{G}}\mathfrak{u}(r) = \mathfrak{u}(r)$,

3. $\mathcal{I}_\mathfrak{a}^{q;\mathbb{G}}(F_\mathfrak{a}(r))^{\ell-1} = \frac{\Gamma(\ell)}{\Gamma(\ell+q)}(F_\mathfrak{a}(r))^{\ell+q-1}$,

4. $\mathcal{D}_\mathfrak{a}^{q;\mathbb{G}}(F_\mathfrak{a}(r))^{\ell-1} = \frac{\Gamma(\ell)}{\Gamma(\ell-q)}(F_\mathfrak{a}(r))^{\ell-q-1}$,

5. $\mathcal{D}_\mathfrak{a}^{q;\mathbb{G}}(F_\mathfrak{a}(r))^k = 0$, for $k \in \{0, \ldots, n-1\}, n \in N, n-1 < q \le n$.

**Lemma 1.4** ([8, 35]). *Let $\alpha \in (n-1, n), \beta > 0, \mathfrak{a} > 0, \mathfrak{u} \in L(\mathfrak{a}, \mathcal{T}), \mathcal{D}_\mathfrak{a}^{\alpha;\mathbb{G}}\mathfrak{u} \in L(\mathfrak{a}, \mathcal{T})$. So*

$$\mathcal{D}_\mathfrak{a}^{\alpha;\mathbb{G}}\mathfrak{u} = 0$$

*has the unique solution* $\mathfrak{u}(r) = \sigma_0 + \sigma_1(\mathbb{G}(r) - \mathbb{G}(\mathfrak{a})) + \sigma_2(\mathbb{G}(r) - \mathbb{G}(\mathfrak{a}))^2 + \cdots + \sigma_{n-1}(\mathbb{G}(r) - \mathbb{G}(\mathfrak{a}))^{n-1}$, *and* $\mathcal{I}_\mathfrak{a}^{\alpha;\mathbb{G}}\mathcal{D}_\mathfrak{a}^{\alpha;\mathbb{G}}\mathfrak{u}(r) = \mathfrak{u}(r) + \sigma_0 + \sigma_1(\mathbb{G}(r) - \mathbb{G}(\mathfrak{a})) + \sigma_2(\mathbb{G}(r) - \mathbb{G}(\mathfrak{a}))^2 + \cdots + \sigma_{n-1}(\mathbb{G}(r) - \mathbb{G}(\mathfrak{a}))^{n-1}$, *with* $\sigma_\ell \in \mathbb{R}, \ell = 0, 1, \ldots, n-1$.

*Besides,*

$$\mathcal{D}_\mathfrak{a}^{\alpha;\mathbb{G}}\mathcal{I}_\mathfrak{a}^{\alpha;\mathbb{G}}\mathfrak{u}(r) = \mathfrak{u}(r),$$

*and*

$$\mathcal{I}_\mathfrak{a}^{\alpha;\mathbb{G}}\mathcal{I}_\mathfrak{a}^{\beta;\mathbb{G}}\mathfrak{u}(r) = \mathcal{I}_\mathfrak{a}^{\beta;\mathbb{G}}\mathcal{I}_\mathfrak{a}^{\alpha;\mathbb{G}}\mathfrak{u}(r) = \mathcal{I}_\mathfrak{a}^{\alpha+\beta;\mathbb{G}}\mathfrak{u}(r).$$

**Lemma 1.5**. *The function $\mathfrak{u} \in C^2([\mathfrak{a} - \tau, \mathfrak{b}], \mathbb{R})$ is a solution of the problem*

$$\mathcal{D}_\mathfrak{a}^{\alpha_1;\mathbb{G}}[\mathcal{D}_\mathfrak{a}^{\alpha_2;\mathbb{G}}\mathfrak{u}(r) - \hbar(r, \mathfrak{u}_r)] = \mathcal{F}(r, \mathfrak{u}_r), \quad r \in J := [\mathfrak{a}, \mathfrak{b}],$$

$$\mathfrak{u}(r) = \phi(r), \quad r \in [\mathfrak{a} - \tau, \mathfrak{a}], \tag{4}$$

$$\mathcal{D}_\mathfrak{a}^{\alpha_2;\mathbb{G}}\mathfrak{u}(\mathfrak{a}) = \eta \in \mathbb{R},$$

*iff*

$$\mathfrak{u}(r) = \begin{cases} \phi(r), & \text{if } r \in [\mathfrak{a} - \tau, \mathfrak{a}], \\[2ex] \phi(\mathfrak{a}) + (\eta - \hbar(\mathfrak{a}, \phi(\mathfrak{a})))\dfrac{(\mathbb{G}(r) - \mathbb{G}(\mathfrak{a}))^{\alpha_2}}{\Gamma(\alpha_2 + 1)} \\[2ex] + \dfrac{1}{\Gamma(\alpha_2)} \displaystyle\int_{\mathfrak{a}}^{r} (\mathbb{G}(r) - \mathbb{G}(s))^{\alpha_2 - 1} \mathbb{G}'(s)\hbar(s, \mathfrak{u}_s)ds \\[2ex] + \dfrac{1}{\Gamma(\alpha_1 + \alpha_2)} \displaystyle\int_{\mathfrak{a}}^{r} (\mathbb{G}(r) - \mathbb{G}(s))^{\alpha_1 + \alpha_2 - 1} \mathbb{G}'(s)\mathcal{F}(s, \mathfrak{u}_s)ds, & \text{if } r \in [\mathfrak{a}, \mathfrak{b}]. \end{cases} \quad (5)$$

*Proof.* The solution of $\mathbb{G}$-Caputo differential Eq in (4) can be written as

$$\mathcal{D}_{\mathfrak{a}}^{\alpha_2;\mathbb{G}}\mathfrak{u}(r) - \hbar(r, \mathfrak{u}_r) = \frac{1}{\Gamma(\alpha_1)} \int_{\mathfrak{a}}^{r} (\mathbb{G}(r) - \mathbb{G}(s))^{\alpha_1 - 1} \mathbb{G}'(s)\mathcal{F}(s, \mathfrak{u}_s)ds + \sigma_1, \quad (6)$$

upon considering the condition $\mathcal{D}_{\mathfrak{a}}^{\alpha_2;\mathbb{G}}\mathfrak{u}(\mathfrak{a}) = \eta$, with $\sigma_1 \in \mathbb{R}$ being an arbitrary constant, we determine that $\sigma_1$ can be expressed as $\sigma_1 = \eta - \hbar(\mathfrak{a}, \phi(\mathfrak{a}))$. Subsequently, we derive the following result:

$$\begin{aligned} \mathfrak{u}(r) &= (\eta - \hbar(\mathfrak{a}, \phi(\mathfrak{a})))\frac{(\mathbb{G}(r) - \mathbb{G}(\mathfrak{a}))^{\alpha_2}}{\Gamma(\alpha_2 + 1)} \\[2ex] &\quad + \frac{1}{\Gamma(\alpha_2)} \int_{\mathfrak{a}}^{r} (\mathbb{G}(r) - \mathbb{G}(s))^{\alpha_2 - 1} \mathbb{G}'(s)\hbar(s, \mathfrak{u}_s)ds \\[2ex] &\quad + \frac{1}{\Gamma(\alpha_1 + \alpha_2)} \int_{\mathfrak{a}}^{r} (\mathbb{G}(r) - \mathbb{G}(s))^{\alpha_1 + \alpha_2 - 1} \mathbb{G}'(s)\mathcal{F}(s, \mathfrak{u}_s)ds + \sigma_2. \end{aligned}$$

The equation above allows us to determine that $\sigma_2 = \phi(\mathfrak{a})$ and the validity of (5) is demonstrated. The reverse can be deduced through straightforward calculations.

## 3 Existence and uniqueness result

Here, we prove the EUS for IVP defined by Eqs (1)–(3).

**Definition 2.1**. *A function $\mathfrak{u} \in C^2([\mathfrak{a} - \tau, \mathfrak{b}], R)$, it is referred to as a solution spanning from* (1)–(3) *if $\mathfrak{u}$ meets the equation $\mathcal{D}_{\mathfrak{a}}^{\alpha_1;\mathbb{G}}[\mathcal{D}_{\mathfrak{a}}^{\alpha_2;\mathbb{G}}\mathfrak{u}(r) - \hbar(r, \mathfrak{u}_r)] = \mathcal{F}(r, \mathfrak{u}_r)$ on $J$, by $\mathfrak{u}(r) = \phi(r)$ on* $[\mathfrak{a} - \tau, \mathfrak{a}]$ *and $\mathcal{D}_{\mathfrak{a}}^{\alpha_2;\mathbb{G}}(\mathfrak{a}) = \eta$.*

The following theorem provides a uniqueness outcome based on the given assumptions.

($\Omega$1) there exists $\ell > 0$ such that

$$|\mathcal{F}(r, \mathcal{X}) - \mathcal{F}(r, \mathcal{Y})| \le \ell\|\mathcal{X} - \mathcal{Y}\|_C, \text{ for } r \in J \text{ and every } \mathcal{X}, \mathcal{Y} \in C_\tau;$$

($\Omega$2) there exists a nonnegative constant $k$ such that

$$|\hbar(r, \mathcal{X}) - \hbar(r, \mathcal{Y})| \le k\|\mathcal{X} - \mathcal{Y}\|_C, \text{ for } r \in J \text{ and every } \mathcal{X}, \mathcal{Y} \in C_\tau.$$

**Theorem 2.2**. *Provided that conditions ($\Omega$1) and ($\Omega$2) are satisfied. If*

$$\frac{k(F_{\mathfrak{a}}(\mathfrak{b}))^{\alpha_2}}{\Gamma(\alpha_2 + 1)} + \frac{\ell(F_{\mathfrak{a}}(\mathfrak{b}))^{\alpha_1 + \alpha_2}}{\Gamma(\alpha_1 + \alpha_2 + 1)} < 1, \quad (7)$$

*in such a case, a unique solution for IVP defined by* Eqs (1)–(3) *exists over the interval* $[\mathfrak{a} - \tau, \mathfrak{b}]$.

*Proof.* Let's examine the operator. $\aleph : C([\mathfrak{a} - \tau, \mathfrak{b}], \mathbb{R}) \to C([\mathfrak{a} - \tau, \mathfrak{b}], \mathbb{R})$ characterized by

$$\aleph(\mathfrak{u})(r) = \begin{cases} \phi(r), & \text{if } r \in [\mathfrak{a} - \tau, \mathfrak{a}], \\[2ex] \phi(\mathfrak{a}) + (\eta - \hbar(\mathfrak{a}, \phi(\mathfrak{a}))) \dfrac{(\mathbb{G}(r) - \mathbb{G}(\mathfrak{a}))^{\alpha_2}}{\Gamma(\alpha_2 + 1)} \\[2ex] + \dfrac{1}{\Gamma(\alpha_2)} \displaystyle\int_{\mathfrak{a}}^{r} (\mathbb{G}(r) - \mathbb{G}(s))^{\alpha_2 - 1} \mathbb{G}'(s) \hbar(s, \mathfrak{u}_s) ds \\[2ex] + \dfrac{1}{\Gamma(\alpha_1 + \alpha_2)} \displaystyle\int_{\mathfrak{a}}^{r} (\mathbb{G}(r) - \mathbb{G}(s))^{\alpha_1 + \alpha_2 - 1} \mathbb{G}'(s) \mathcal{F}(s, \mathfrak{u}_s) ds, & \text{if } r \in J. \end{cases} \tag{8}$$

In order to demonstrate that the operator $\aleph$ is a contraction, consider $\mathfrak{u}, z \in C([\mathfrak{a} - \tau, \mathfrak{b}], \mathbb{R})$. Then, we can observe the following:

$$\begin{aligned} |\aleph(\mathfrak{u})(r) - \aleph(z)(r)| \quad &\leq \frac{1}{\Gamma(\alpha_2)} \int_{\mathfrak{a}}^{r} F_s(r)^{\alpha_2 - 1} \mathbb{G}'(s) |\hbar(s, \mathfrak{u}_s) - \hbar(s, z_s)| ds \\[1ex] &\quad + \frac{1}{\Gamma(\alpha_1 + \alpha_2)} \int_{\mathfrak{a}}^{r} F_s(r)^{\alpha_1 + \alpha_2 - 1} \mathbb{G}'(s) |\mathcal{F}(s, \mathfrak{u}_s) - \mathcal{F}(s, z_s)| ds \\[1ex] &\leq \frac{k}{\Gamma(\alpha_2)} \int_{\mathfrak{a}}^{r} F_s(r)^{\alpha_2 - 1} \mathbb{G}'(s) \|\mathfrak{u}_s - z_s\|_C ds \\[1ex] &\quad + \frac{\ell}{\Gamma(\alpha_1 + \alpha_2)} \int_{\mathfrak{a}}^{r} F_s(r)^{\alpha_1 + \alpha_2 - 1} \mathbb{G}'(s) \|\mathfrak{u}_s - z_s\|_C ds \\[1ex] &\leq \frac{k(F_{\mathfrak{a}}(r))^{\alpha_2}}{\Gamma(\alpha_2 + 1)} \|\mathfrak{u} - z\|_{[\mathfrak{a} - \tau, \mathfrak{b}]} + \frac{\ell(F_{\mathfrak{a}}(r))^{\alpha_1 + \alpha_2}}{\Gamma(\alpha_1 + \alpha_2 + 1)} \|\mathfrak{u} - z\|_{[\mathfrak{a} - \tau, \mathfrak{b}]}. \end{aligned}$$

Consequently we obtain

$$\|\aleph(\mathfrak{u}) - \aleph(z)\|_{[\mathfrak{a} - \tau, \mathfrak{b}]} \leq \frac{k(F_{\mathfrak{a}}(\mathfrak{b}))^{\alpha_2}}{\Gamma(\alpha_2 + 1)} + \frac{\ell(F_{\mathfrak{a}}(\mathfrak{b}))^{\alpha_1 + \alpha_2}}{\Gamma(\alpha_1 + \alpha_2 + 1)} \|\mathfrak{u} - z\|_{[\mathfrak{a} - \tau, \mathfrak{b}]}.$$

Considering (7), this implies the contraction of the operator $\aleph$. Consequently, by Banach's contraction principle, $\aleph$ possesses a unique fixed point. Thus, demonstrating the EUS to the problem (1)–(3) over the interval $[\mathfrak{a} - \tau, \mathfrak{b}]$.

## 4 Existence results

This section is devoted to the presentation of the findings regarding the existence of solutions for the initial value problem outlined in Eqs (1)–(3). The next result is based on the nonlinear version of the Leray-Schauder theorem.

**Lemma 3.1** ([34]). *Consider E as a Banach space, C as a closed and convex subset of C, and U as an open subset of C with the inclusion of $0 \in U$. Assuming that R is a continuous and compact mapping (meaning $R(\bar{U})$ is a subset of C that is relatively compact), then either:*

*(i) R has a fixed point in $\bar{U}$, or*

*(ii) there is a $\mathcal{X} \in \partial U$ (the boundary of U in C) and $\lambda \in (0, 1)$ with $\mathcal{X} = \lambda R(\mathcal{X})$.*

The following assumptions are required for the upcoming theorem:

($\Omega$3) There are continuous functions $\mathcal{F}, \hbar : J \times C_\tau \to \mathbb{R}$;

($\Omega$4) There is a continuous nondecreasing function $H : [0, \infty) \to (0, \infty)$ and a function $p \in C(J, \mathbb{R}^+)$ satisfying

$$|\mathcal{F}(r, \mathcal{X})| \leq p(r) H(\|\mathcal{X}\|_C) \ \forall \ (r, \mathcal{X}) \in J \times C_\tau.$$

($\Omega$5) There exist constants $d_1 < \Gamma(\alpha_2 + 1)(F_a(\mathfrak{b}))^{-\alpha_2}$ and $d_2 \geq 0$ such that

$$|\hbar(r, \mathcal{X})| \leq d_1 \|\mathcal{X}\|_C + d_2, \ r \in J, \ \mathcal{X} \in C_\tau.$$

($\Omega$6) There exists a constant $\mathcal{M} > 0$ such that

$$\frac{1 - \dfrac{d_1 (F_a(\mathfrak{b}))^{\alpha_2}}{\Gamma(\alpha_2 + 1)} \mathcal{M}}{\mathcal{M}_0 + H(\mathcal{M}) \|p\|_\infty \dfrac{1}{\Gamma(\alpha_1 + \alpha_2 + 1)} (F_a(\mathfrak{b}))^{\alpha_1 + \alpha_2}} > 1,$$

where

$$\mathcal{M}_0 = \|\phi\|_C + |\eta| + d_1 \|\phi\|_C + 2 d_2 \frac{(F_a(\mathfrak{b}))^{\alpha_2}}{\Gamma(\alpha_2 + 1)}.$$

**Theorem 3.2**. *With the suppositions* ($\Omega$3)–($\Omega$6) *hold, IVP* (1)–(3) *possesses a solution within* $[\mathfrak{a} - \tau, \mathfrak{b}]$.

*Proof.* We will demonstrate that $\aleph : C([\mathfrak{a} - \tau, \mathfrak{b}], \mathbb{R}) \to C([\mathfrak{a} - \tau, \mathfrak{b}], \mathbb{R})$ is continuous and completely continuous.

**Step 1**: $\aleph$ is continuously defined. Consider a sequence $\{\mathfrak{u}_n\}$ such that $\mathfrak{u}_n \to \mathfrak{u}$ in $C([\mathfrak{a} - \tau, \mathfrak{b}], \mathbb{R})$. So

$$|\aleph(\mathfrak{u}_n)(r) - \aleph(\mathfrak{u})(r)|$$

$$\leq \frac{1}{\Gamma(\alpha_2)} \int_\mathfrak{a}^r F_s(r)^{\alpha_2 - 1} \mathbb{G}'(s) |\hbar(s, \mathfrak{u}_{ns}) - \hbar(s, \mathfrak{u}_s)| ds$$

$$+ \frac{1}{\Gamma(\alpha_1 + \alpha_2)} \int_\mathfrak{a}^r F_s(r)^{\alpha_1 + \alpha_2 - 1} \mathbb{G}'(s) |\mathcal{F}(s, \mathfrak{u}_{ns}) - \mathcal{F}(s, \mathfrak{u}_s)| ds$$

$$\leq \frac{1}{\Gamma(\alpha_2)} \int_\mathfrak{a}^\mathfrak{b} F_s(r)^{\alpha_2 - 1} \mathbb{G}'(s) \sup_{s \in [\mathfrak{a}, \mathfrak{b}]} |\hbar(s, \mathfrak{u}_{ns}) - \hbar(s, \mathfrak{u}_s)| ds$$

$$+ \frac{1}{\Gamma(\alpha_1 + \alpha_2)} \int_\mathfrak{a}^\mathfrak{b} F_s(r)^{\alpha_1 + \alpha_2 - 1} \mathbb{G}'(s) \sup_{s \in [\mathfrak{a}, \mathfrak{b}]} |\mathcal{F}(s, \mathfrak{u}_{ns}) - \mathcal{F}(s, \mathfrak{u}_s)| ds$$

$$\leq \frac{\|\hbar(\cdot, \mathfrak{u}_{n.}) - \hbar(\cdot, \mathfrak{u}_.)\|_\infty}{\Gamma(\alpha)} \int_\mathfrak{a}^\mathfrak{b} F_s(r)^{\alpha_2 - 1} \mathbb{G}'(s) ds$$

$$+ \frac{\|\mathcal{F}(\cdot, \mathfrak{u}_{n.}) - \mathcal{F}(\cdot, \mathfrak{u}_.)\|_\infty}{\Gamma(\alpha_1 + \alpha_2)} \int_\mathfrak{a}^\mathfrak{b} F_s(r)^{\alpha + \beta - 1} \mathbb{G}'(s) ds$$

$$\leq \frac{(F_a(\mathfrak{b}))^{\alpha_2} \|\hbar(\cdot, \mathfrak{u}_{n.}) - \hbar(\cdot, \mathfrak{u}_.)\|_\infty}{\Gamma(\alpha_2 + 1)}$$

$$+ \frac{(F_a(\mathfrak{b}))^{\alpha_1 + \alpha_2} \|\mathcal{F}(\cdot, \mathfrak{u}_{n.}) - \mathcal{F}(\cdot, \mathfrak{u}_.)\|_\infty}{\Gamma(\alpha_1 + \alpha_2 + 1)}.$$

Because both $\mathcal{F}$ and $\hbar$ are continuous functions, we obtain

$$\|\aleph(\mathfrak{u}_n) - \aleph(\mathfrak{u})\|_\infty$$
$$\leq \frac{(F_\mathfrak{a}(\mathfrak{b}))^{\alpha_2}\|\hbar(\cdot, \mathfrak{u}_{n.}) - \hbar(\cdot, \mathfrak{u}_.)\|_\infty}{\Gamma(\alpha_2 + 1)} + \frac{(F_\mathfrak{a}(\mathfrak{b}))^{\alpha_1 + \alpha_2}\|\mathcal{F}(\cdot, \mathfrak{u}_{n.}) - \mathcal{F}(\cdot, \mathfrak{u}_.)\|_\infty}{\Gamma(\alpha_1 + \alpha_2 + 1)} \to 0$$

as $n \to \infty$.

**Step 2**: The operator $\aleph$ transforms bounded sets into bounded sets within the function space $C([\mathfrak{a} - \tau, \mathfrak{b}], \mathbb{R})$. Sufficiently to prove for any $\theta > 0$ there exists $\tilde{\ell} > 0$ such that for each $\mathfrak{u} \in \mathcal{Q}_\theta = \{\mathfrak{u} \in C([\mathfrak{a} - \tau, \mathfrak{b}], \mathbb{R}) : \|\mathfrak{u}\|_\infty \leq \theta\}$, we have $\|\aleph(\mathfrak{u})\|_\infty \leq \tilde{\ell}$. By ($\Omega4$) and ($\Omega5$), for each $r \in J$, we get

$$\begin{aligned}
|\aleph(\mathfrak{u})(r)| \quad &\leq \|\phi\|_C + |\eta| + d_1\|\phi\|_C + d_2\frac{(F_\mathfrak{a}(\mathfrak{b}))^{\alpha_2}}{\Gamma(\alpha_2 + 1)} \\
&\quad + \frac{1}{\Gamma(\alpha_2)}\int_\mathfrak{a}^r F_s(r)^{\alpha_2 - 1}\mathbb{G}'(s)|\hbar(s, \mathfrak{u}_s)|ds \\
&\quad + \frac{1}{\Gamma(\alpha_1 + \alpha_2)}\int_\mathfrak{a}^r (F_s(r)^{\alpha_1 + \alpha_2 - 1}\mathbb{G}'(s)|\mathcal{F}(s, \mathfrak{u}_s)|ds \\
&\leq \|\phi\|_C + |\eta| + d_1\|\phi\|_C + d_2\frac{(F_\mathfrak{a}(\mathfrak{b}))^{\alpha_2}}{\Gamma(\alpha_2 + 1)} \\
&\quad + \frac{d_1\|\mathfrak{u}\|_{[\mathfrak{a}-\tau, \mathfrak{b}]} + d_2}{\Gamma(\alpha_2)}\int_\mathfrak{a}^r F_s(r)^{\alpha_2 - 1}\mathbb{G}'(s)ds \\
&\quad + \frac{H(\|\mathfrak{u}\|_{[\mathfrak{a}-\tau, \mathfrak{b}]})\|p\|_\infty}{\Gamma(\alpha_1 + \alpha_2)}\int_\mathfrak{a}^r F_s(r)^{\alpha_1 + \alpha_2 - 1}\mathbb{G}'(s)ds \\
&\leq \|\phi\|_C + |\eta| + d_1\|\phi\|_C + d_2\frac{(F_\mathfrak{a}(\mathfrak{b}))^{\alpha_2}}{\Gamma(\alpha_2 + 1)} \\
&\quad + \frac{d_1\|\mathfrak{u}\|_{[\mathfrak{a}-\tau, \mathfrak{b}]} + d_2}{\Gamma(\alpha_2 + 1)}(F_\mathfrak{a}(\mathfrak{b}))^{\alpha_2} + \frac{H(\|\mathfrak{u}\|_{[\mathfrak{a}-\tau, \mathfrak{b}]})\|p\|_\infty}{\Gamma(\alpha_1 + \alpha_2 + 1)}(F_\mathfrak{a}(\mathfrak{b}))^{\alpha_1 + \alpha_2}.
\end{aligned}$$

Thus

$$\|\aleph(\mathfrak{u})\|_\infty \leq \|\phi\|_C + \left[|\eta| + d_1(\|\phi\|_C + \theta) + 2d_2\right]\frac{(F_\mathfrak{a}(\mathfrak{b}))^{\alpha_2}}{\Gamma(\alpha_2 + 1)}$$
$$+ \frac{H(\theta)\|p\|_\infty}{\Gamma(\alpha_1 + \alpha_2 + 1)}(F_\mathfrak{a}(\mathfrak{b}))^{\alpha_1 + \alpha_2} := \tilde{\ell}.$$

**Step 3**: Now we show that the image of a bounded set by $\aleph$ is an equicontinuous subset of $C([\mathfrak{a} - \tau, \mathfrak{b}], \mathbb{R})$. Indeed; suppose that $r_1, r_2 \in J$, $r_1 < r_2$, $\mathcal{Q}_\theta$ is a bounded subset of

$C([\mathfrak{a} - \tau, \mathfrak{b}], \mathbb{R})$, and take $\mathfrak{u} \in \mathcal{Q}_\theta$. We have

$$
\begin{aligned}
&|\aleph(\mathfrak{u})(r_2) - \aleph(\mathfrak{u})(r_1)| \\
&\leq \frac{|\eta| + d_1\|\phi\|_C + d_2}{\Gamma(\alpha_2 + 1)}(F_\mathfrak{a}(r_2))^{\alpha_2} - (F_\mathfrak{a}(r_1))^{\alpha_2} \\
&\quad + \Big|\frac{1}{\Gamma(\alpha_2)}\int_\mathfrak{a}^{r_1}(F_s(r_2)^{\alpha_2 - 1} - F_s(r_1)^{\alpha_2 - 1})\mathbb{G}'(s)\hbar(s, \mathfrak{u}_s)ds \\
&\quad + \frac{1}{\Gamma(\alpha_2)}\int_{r_1}^{r_2}(F_s(r_2))^{\alpha_2 - 1}\mathbb{G}'(s)\hbar(s, \mathfrak{u}_s)ds\Big| \\
&\quad + \Big|\frac{1}{\Gamma(\alpha_1 + \alpha_2)}\int_\mathfrak{a}^{r_1}\big[(F_s(r_2))^{\alpha_1 + \alpha_2 - 1} - (F_s(r_1))^{\alpha_1 + \alpha_2 - 1}\big]\mathbb{G}'(s)\mathcal{F}(s, \mathfrak{u}_s)ds \\
&\quad + \frac{1}{\Gamma(\alpha_1 + \alpha_2)}\int_{r_1}^{r_2}(F_s(r_2))^{\alpha_1 + \alpha_2 - 1}\mathbb{G}'(s)\mathcal{F}(s, \mathfrak{u}_s)ds\Big| \\
&\leq \frac{|\eta| + d_1\|\phi\|_C + d_2}{\Gamma(\alpha_2 + 1)}\big[(F_\mathfrak{a}(r_2))^{\alpha_2} - (F_\mathfrak{a}(r_1))^{\alpha_2}\big] \\
&\quad + \frac{d_1\theta + d_2}{\Gamma(\alpha_2)}\int_\mathfrak{a}^{r_1}\big[(F_s(r_2))^{\alpha_2 - 1} - (F_s(r_1))^{\alpha_2 - 1}\big]\mathbb{G}'(s)ds \\
&\quad + \frac{d_1\theta + d_2}{\Gamma(\alpha_2)}\int_{r_1}^{r_2}(F_s(r_2))^{\alpha_2 - 1}\mathbb{G}'(s)ds \\
&\quad + \frac{H(\theta)\|p\|_\infty}{\Gamma(\alpha_1 + \alpha_2)}\int_\mathfrak{a}^{r_1}\big[(F_s(r_2))^{\alpha_1 + \alpha_2 - 1} - (F_s(r_1))^{\alpha_1 + \alpha_2 - 1}\big]\mathbb{G}'(s)ds \\
&\quad + \frac{H(\theta)\|p\|_\infty}{\Gamma(\alpha_1 + \alpha_2)}\int_{r_1}^{r_2}(F_s(r_2))^{\alpha_1 + \alpha_2 - 1}\mathbb{G}'(s)ds \\
&\leq \frac{|\eta| + d_1\|\phi\|_C + d_2}{\Gamma(\alpha_2 + 1)}\big[(F_\mathfrak{a}(r_2))^{\alpha_2} - (F_\mathfrak{a}(r_1))^{\alpha_2}\big] \\
&\quad + \frac{d_1\theta + d_2}{\Gamma(\alpha_2 + 1)}\Big[|(F_\mathfrak{a}(r_2))^{\alpha_2} - (F_\mathfrak{a}(r_1))^{\alpha_2}| + |F_{r_1}(r_2)|^{\alpha_2}\Big] \\
&\quad + \frac{H(\theta)\|p\|_\infty}{\Gamma(\alpha_1 + \alpha_2 + 1)}\Big[|(F_\mathfrak{a}(r_2))^{\alpha_1 + \alpha_2} - (F_\mathfrak{a}(r_1))^{\alpha_1 + \alpha_2}| + |F_{r_1}(r_2)|^{\alpha_1 + \alpha_2}\Big].
\end{aligned}
$$

As $r_1$ approaches $r_2$, the right-hand side of the mentioned inequality approaches zero. Equi-continuity in cases where $r_1$ is less than $r_2$ and when $r_1$ is less than or equal to zero while $r_2$ is greater than or equal to zero is readily apparent.

As a result of Steps 1 to 3, applying the theorem of Arzelá-Ascoli leads to the fact that $\aleph : C([\mathfrak{a} - \tau, \mathfrak{b}], \mathbb{R}) \to C([\mathfrak{a} - \tau, \mathfrak{b}], \mathbb{R})$, is continuous as well as completely continuous.

**Step 4**: We demonstrate the existence of an open set $U \subseteq C([\mathfrak{a} - \tau, \mathfrak{b}], \mathbb{R})$ with $\mathfrak{u} \neq \lambda\aleph(\mathfrak{u})$ for $\lambda \in (0, 1)$ and $\mathfrak{u} \in \partial U$. Let $\mathfrak{u} \in C([\mathfrak{a} - \tau, \mathfrak{b}], \mathbb{R})$ and $\mathfrak{u} = \lambda\aleph(\mathfrak{u})$ for some $0 < \lambda < 1$. Then, for each $r \in J$, we have

$$
\begin{aligned}
\mathfrak{u}(r) &= \lambda(\phi(\mathfrak{a}) + (\eta - \hbar(\mathfrak{a}, \phi(\mathfrak{a})))\frac{(F_\mathfrak{a}(\mathfrak{b}))^{\alpha_2}}{\Gamma(\alpha_2 + 1)} + \frac{1}{\Gamma(\alpha_2)}\int_\mathfrak{a}^r(F_s(r))^{\alpha_2 - 1}\mathbb{G}'(s)\hbar(s, \mathfrak{u}_s)ds \\
&\quad + \frac{1}{\Gamma(\alpha_1 + \alpha_2)}\int_\mathfrak{a}^r(F_s(r))^{\alpha_1 + \alpha_2 - 1}\mathcal{F}(s, \mathfrak{u}_s)ds).
\end{aligned}
$$

Based on the assumptions we have made, for every $r \in J$, we achieve

$$
\begin{aligned}
|\mathfrak{u}(r)| \quad & \leq \|\phi\|_C + \left[|\eta| + d_1\|\phi\|_C + d_2\right] \frac{(F_\mathfrak{a}(\mathfrak{b}))^{\alpha_2}}{\Gamma(\alpha_2 + 1)} \\
& + \frac{d_1\|\mathfrak{u}\|_{[\mathfrak{a}-\tau,\mathfrak{b}]} + d_2}{\Gamma(\alpha_2)} \int_\mathfrak{a}^r (F_s(r))^{\alpha_2-1} \mathbb{G}'(s)ds \\
& + \frac{1}{\Gamma(\alpha_1 + \alpha_2)} \int_\mathfrak{a}^r (F_s(r))^{\alpha_1+\alpha_2-1} p(s)H(\|\mathfrak{u}_s\|_C)\mathbb{G}'(s)ds \\
& \leq \|\phi\|_C + \left[|\eta| + d_1\|\phi\|_C + d_2\right] \frac{(F_\mathfrak{a}(\mathfrak{b}))^{\alpha_2}}{\Gamma(\alpha_2 + 1)} + \left[d_1\|\mathfrak{u}\|_{[\mathfrak{a}-\tau,\mathfrak{b}]} + d_2\right] \left(\frac{(F_\mathfrak{a}(\mathfrak{b}))^{\alpha_2}}{\Gamma(\alpha_2 + 1)}\right. \\
& + \frac{\|p\|_\infty H(\|\mathfrak{u}\|_{[\mathfrak{a}-\tau,\mathfrak{b}]})}{\Gamma(\alpha_1 + \alpha_2 + 1)} (F_\mathfrak{a}(\mathfrak{b}))^{\alpha_1+\alpha_2},
\end{aligned}
$$

we can also write it as follows

$$
\frac{\left(1 - \frac{d_1(F_\mathfrak{a}(\mathfrak{b}))^{\alpha_2}}{\Gamma(\alpha_2 + 1)}\right)\|\mathfrak{u}\|_{[\mathfrak{a}-\tau,\mathfrak{b}]}}{\mathcal{M}_0 + H(\|\mathfrak{u}\|_{[\mathfrak{a}-\tau,\mathfrak{b}]})\|p\|_\infty \frac{1}{\Gamma(\alpha_1 + \alpha_2 + 1)} (F_\mathfrak{a}(\mathfrak{b}))^{\alpha_1+\alpha_2}} \leq 1.
$$

In view of ($\Omega$6), there exists $\mathcal{M}$ such that $\|\mathfrak{u}\|_{[\mathfrak{a}-\tau,\mathfrak{b}]} \neq \mathcal{M}$. Let us set

$$
U = \{\mathfrak{u} \in C([\mathfrak{a} - \tau, \mathfrak{b}], \mathbb{R}) : \|\mathfrak{u}\|_{[\mathfrak{a}-\tau,\mathfrak{b}]} < \mathcal{M}\}.
$$

It's worth noting that the operator $\aleph : \bar{U} \to C([\mathfrak{a} - \tau, \mathfrak{b}], \mathbb{R})$, exhibits both continuity and complete continuity. Given the selection of $U$, there exists no $\mathfrak{u}$ on the boundary $\partial U$ such that $\mathfrak{u} = \lambda\aleph\mathfrak{u}$ for any $\lambda$ in $(0, 1)$. Consequently, employing the nonlinear alternative of Leray-Schauder type (Lemma 3.1), we conclude that $\aleph$ possesses a fixed point $\mathfrak{u} \in \bar{U}$, which serves as a solution to the problem described in (1)–(3). This successfully concludes the proof.

**Lemma 3.3** (KFPT [33]). *Consider S as a closed, bounded, convex, and non-empty subset of a Banach space X. Operators $\mathcal{P}$ and $\mathcal{Q}$ are defined as follows:*

- *$\mathcal{P}y + \mathcal{Q}x \in S$ whenever $y, \mathfrak{u} \in S$;*

- *$\mathcal{P}$ is continuous and compact;*

- *$\mathcal{Q}$ is a contraction.*

*Then there exists $z \in S$ so that $z = \mathcal{P}z + \mathcal{Q}z$.*

**Theorem 3.4**. *Assume that the conditions ($\Omega$2) and ($\Omega$3) are satisfied and that*
*($\Omega$7) $|\mathcal{F}(r, x)| \leq \mu(r), |\hbar(r, x)| \leq \nu(r)$, for all $(r, x) \in J \times \mathbb{R}$, and $\mu, \nu \in C(J, \mathbb{R}^+)$.*
*Then we have at least one solution of problem (1)–(3) on $[\mathfrak{a} - \tau, \mathfrak{b}]$, if*

$$
\frac{k(F_\mathfrak{a}(\mathfrak{b}))^{\alpha_2}}{\Gamma(\alpha_2 + 1)} < 1. \tag{9}
$$

*Proof.* We define the operators $\Xi_1$ and $\Xi_2$ by

$$
\Xi_1\mathfrak{u}(r) = \begin{cases} 0, & \text{if } r \in [\mathfrak{a}-\tau, \mathfrak{a}], \\[2mm] (\eta - \hbar(\mathfrak{a}, \phi(\mathfrak{a}))) \dfrac{(F_\mathfrak{a}(\mathfrak{b}))^{\alpha_2}}{\Gamma(\alpha_2 + 1)} \\[2mm] \quad + \dfrac{1}{\Gamma(\alpha_2)} \displaystyle\int_\mathfrak{a}^r (F_s(r))^{\alpha_2 - 1} \mathbb{G}'(s) \hbar(s, \mathfrak{u}_s) ds, & \text{if } r \in J. \end{cases} \tag{10}
$$

$$
\Xi_2\mathfrak{u}(r) = \begin{cases} \phi(r), & \text{if } r \in [\mathfrak{a}-\tau, \mathfrak{a}], \\[2mm] \phi(\mathfrak{a}) + \dfrac{1}{\Gamma(\alpha_1 + \alpha_2)} \displaystyle\int_\mathfrak{a}^r (F_s(r))^{\alpha_1 + \alpha_2 - 1} \mathbb{G}'(s) \mathcal{F}(s, \mathfrak{u}_s) ds, & \text{if } r \in J. \end{cases} \tag{11}
$$

Setting $\sup_{r \in [\mathfrak{a},\mathfrak{b}]} \mu(r) = \|\mu\|_\infty$, $\sup_{r \in [\mathfrak{a},\mathfrak{b}]} v(r) = \|v\|_\infty$ and choosing

$$
\rho \geq \|\phi\|_C + [|\eta| + 2\|v\|_\infty] \frac{(F_\mathfrak{a}(\mathfrak{b}))^{\alpha_2}}{\Gamma(\alpha_2 + 1)} + \|\mu\|_\infty \frac{(F_\mathfrak{a}(\mathfrak{b}))^{\alpha_1 + \alpha_2}}{\Gamma(\alpha_1 + \alpha_2 + 1)}, \tag{12}
$$

we consider $\mathcal{Q}_\rho = \{\mathfrak{u} \in C([\mathfrak{a}-\tau, \mathfrak{b}], \mathbb{R}) : \|\mathfrak{u}\|_\infty \leq \rho\}$. For any $\mathfrak{u}, z \in \mathcal{Q}_\rho$, we have

$$
\begin{aligned}
&|\Xi_1\mathfrak{u}(r) + \Xi_2 z(r)| \\[2mm]
&\leq \sup_{r \in [\mathfrak{a},\mathfrak{b}]} \Big\{ (\eta - \hbar(\mathfrak{a}, \phi)) \frac{(F_\mathfrak{a}(r))^{\alpha_2}}{\Gamma(\alpha_2 + 1)} + \frac{1}{\Gamma(\alpha_2)} \int_\mathfrak{a}^r (F_s(r))^{\alpha_2 - 1} \mathbb{G}'(s) \hbar(s, \mathfrak{u}_s) ds \\[2mm]
&\qquad + \phi(\mathfrak{a}) + \frac{1}{\Gamma(\alpha_1 + \alpha_2)} \int_\mathfrak{a}^r (F_s(r))^{\alpha_1 + \alpha_2 - 1} \mathbb{G}'(s) \mathcal{F}(s, \mathfrak{u}_s) ds \Big\} \\[2mm]
&\leq \|\phi\|_C + [|\eta| + 2\|v\|_\infty] \frac{(F_\mathfrak{a}(\mathfrak{b}))^{\alpha_2}}{\Gamma(\alpha_2 + 1)} + \|\mu\|_\infty \frac{(F_\mathfrak{a}(\mathfrak{b}))^{\alpha_1 + \alpha_2}}{\Gamma(\alpha_1 + \alpha_2 + 1)} \\[2mm]
&\leq \rho.
\end{aligned}
$$

This shows that $\Xi_1\mathfrak{u} + \Xi_2 z \in \mathcal{Q}_\rho$. Using (9) one can deduce that $\Xi_1$ is a contraction.

Since $\mathcal{F}$ is continuous, the operator $\Xi_2$ is continuous. Additionally, $\Xi_2$ is uniformly bounded on $\mathcal{Q}_\rho$ by

$$
\|\Xi_2\mathfrak{u}\| \leq \|\phi\|_C + \|\mu\|_\infty \frac{(F_\mathfrak{a}(\mathfrak{b}))^{\alpha_1 + \alpha_2}}{\Gamma(\alpha_1 + \alpha_2 + 1)}.
$$

Next, we will establish the compactness of the operator $\Xi_2$. To do this, we introduce the following definitions

$$
\mathcal{F} = \sup_{(r,\mathfrak{u}) \in [\mathfrak{a},\mathfrak{b}] \times \mathcal{Q}_\rho} |\mathcal{F}(r, \mathfrak{u})| < \infty,
$$

and consequently, for $r_1, r_2 \in [\mathfrak{a}, \mathfrak{b}], r_1 < r_2$, we have

$$
\begin{aligned}
&|\Xi_2 \mathfrak{u}(r_2) - \Xi_2 \mathfrak{u}(r_1)| \\
&\leq \frac{\mathcal{F}}{\Gamma(\alpha_1 + \alpha_2)} \int_{\mathfrak{a}}^{r_1} \left| (F_s(r_2))^{\alpha_1 + \alpha_2 - 1} - (F_s(r_1))^{\alpha_1 + \alpha_2 - 1} \right| ds \\
&\quad + \frac{\mathcal{F}}{\Gamma(\alpha_1 + \alpha_2)} \int_{r_1}^{r_2} (F_s(r_2))^{\alpha_1 + \alpha_2 - 1} ds \\
&\leq \frac{\mathcal{F}}{\Gamma(\alpha_1 + \alpha_2 + 1)} \left[ |(F_{\mathfrak{a}}(r_2))^{\alpha_1 + \alpha_2} - (F_{\mathfrak{a}}(r_1))^{\alpha_1 + \alpha_2}| + |F_{r_1}(r_2)|^{\alpha_1 + \alpha_2} \right],
\end{aligned}
$$

which is independent of $\mathfrak{u}$ and tends to zero as $r_2 - r_1 \to 0$. Thus, $\Xi_2$ is equicontinuous. So $\Xi_2$ is relatively compact on $\mathcal{Q}_\rho$. Hence, by the Arzelá-Ascoli theorem, $\Xi_2$ is compact on $\mathcal{Q}_\rho$. Thus all the assumptions of Lemma 3.3 are satisfied. So the conclusion of Lemma 3.3 implies that the problem (1)–(3) has at least one solution on $[\mathfrak{a} - \tau, \mathfrak{b}]$

## 5 Initial value integral

The findings presented in this work can be expanded to encompass scenarios involving an initial value integral condition structured as follows:

$$
\mathcal{D}_{\mathfrak{a}}^{\alpha_2, \mathbb{G}} \mathfrak{u}(\mathfrak{a}) = \int_{\mathfrak{a}}^{\mathfrak{b}} W(s, \mathfrak{u}_s) ds, \tag{13}
$$

In the context where $W : J \times C([-\tau, 0], \mathbb{R}) \to \mathbb{R}$, the variable $\eta$ within Eq (8) will be substituted with $\int_{\mathfrak{a}}^{\mathfrak{b}} W(s, \mathfrak{u}_s) ds$. As a consequence, we can express the existence and uniqueness statement for the problem encompassing Eqs (1) and (2)–(13) in the following manner.

**Theorem 4.1**. *Given the fulfillment of conditions* $(\Omega 1)$ *and* $(\Omega 2)$, *we also make the additional assumption that*

$(\Omega 8)$ *there exists a nonnegative constant m such that*

$$
|W(r, \mathcal{X}) - W(r, \mathcal{Y})| \leq m \|\mathcal{X} - \mathcal{Y}\|_C, \text{ for } r \in J \text{ and any } \mathcal{X}, \mathcal{Y} \in C_\tau.
$$

*Hence the problem* (1) *and* (2)–(13) *has a unique solution on* $[\mathfrak{a} - \tau, \mathfrak{b}]$ *if*

$$
\frac{[m(\mathfrak{b} - 1) + k](F_{\mathfrak{a}}(\mathfrak{b}))^{\alpha_2}}{\Gamma(\alpha_2 + 1)} + \frac{\ell(F_{\mathfrak{a}}(\mathfrak{b}))^{\alpha_1 + \alpha_2}}{\Gamma(\alpha_1 + \alpha_2 + 1)} < 1.
$$

The proof of the aforementioned theorem closely resembles that of Theorem 2.2.

The analogous form of the existence results, as seen in Theorems 3.2 and 3.4, can be developed for the problem described by Eqs (1) and (2)–(13) in a comparable fashion.

## 6 Example

In this section, we provide an example to demonstrate how our primary findings can be applied effectively. We will examine a fractional functional differential equation to illustrate

this concept, let $\mathbb{G}(r) = \log r$

$$\mathcal{D}_1^{\frac{2}{5};\mathbb{G}} \mathcal{D}_1^{\frac{1}{5};\mathbb{G}} \mathfrak{u}(r) - \frac{e^{-r}}{\sqrt{5}} \frac{\|\mathfrak{u}_r\|_C}{(1 + \|\mathfrak{u}_r\|_C)} = \frac{\|\mathfrak{u}_r\|_C}{\sqrt{7}(1 + \|\mathfrak{u}_r\|_C)} + r, \tag{14}$$

$$r \in J := [1, e],$$

$$\mathfrak{u}(r) = \phi(r), \ \ r \in [1 - \tau, 1], \tag{15}$$

$$\mathcal{D}_1^{\frac{1}{5};\mathbb{G}} \mathfrak{u}(1) = 1/2. \tag{16}$$

Let

$$\mathcal{F}(r, x) = \frac{x}{\sqrt{7}(1 + x)} + r, \ \ \hbar(r, x) = \frac{e^{-r}}{\sqrt{5}} \frac{x}{1 + x}, \ \ (r, x) \in [1, e] \times [0, \infty).$$

For $y, \mathfrak{u} \in [0, \infty)$ and $r \in J$, we have

$$|\mathcal{F}(r, y) - \mathcal{F}(r, \mathfrak{u})| = \frac{1}{\sqrt{7}} \left| \frac{y}{1 + y} - \frac{\mathfrak{u}}{1 + \mathfrak{u}} \right| = \frac{|y - \mathfrak{u}|}{\sqrt{7}(1 + y)(1 + \mathfrak{u})} \leq \frac{1}{\sqrt{7}} |y - \mathfrak{u}|,$$

and

$$|\hbar(r, y) - \hbar(r, \mathfrak{u})| = \frac{e^{-r}}{\sqrt{5}} \left| \frac{y}{1 + y} - \frac{\mathfrak{u}}{1 + \mathfrak{u}} \right| = \frac{e^{-r}}{\sqrt{5}} \frac{|y - \mathfrak{u}|}{(1 + y)(1 + \mathfrak{u})}$$

$$\leq \frac{1}{\sqrt{5}} |y - \mathfrak{u}|.$$

Therefore, conditions ($\Omega 1$) and ($\Omega 2$) are satisfied with $\ell = \frac{1}{\sqrt{7}}$ and $k = \frac{1}{\sqrt{5}}$ respectively. Since $\frac{k(F_1(\mathfrak{b}))^{\alpha_2}}{\Gamma(\alpha_2 + 1)} + \frac{\ell(F_1(\mathfrak{b}))^{\alpha_1 + \alpha_2}}{\Gamma(\alpha_1 + \alpha_2 + 1)} \approx 0.910079666 < 1$, hence as asserted by Theorem 2.2, the problem encompassing Eqs (14)–(16) possesses a sole, distinct solution within the interval $[1 - \tau, e]$.

Also $|\mathcal{F}(r, x)| \leq \frac{1}{\sqrt{7}} + r = \mu(r)$, $|\hbar(r, x)| \leq \frac{1}{\sqrt{5}} e^{-r} = v(r)$ and $k \frac{(F_1(\mathfrak{b}))^{\alpha_2}}{\Gamma(\alpha_2 + 1)} \approx 0.487071271 < 1$. The conditions stated in Theorem 3.4 are evidently met. As a direct consequence of the theorem's conclusion, a solution to the problem described in Eqs (14)–(16) is guaranteed to exist within the interval $[1 - \tau, e]$.

## 7 Conclusion

In our research, we touched on the theory of existence and uniqueness of a kind of complex equations included neutral functional differential equations by attracting the generalized fractional derivation $\mathbb{G}$-Caputo derivative.

## Acknowledgments

This research has been funded by Scientific Research Deanship at University of Ha'il—Saudi Arabia through project number RG-23 036.

## Author Contributions

**Conceptualization:** Rabah Debbar, Hamid Boulares, Abdelkader Moumen.

**Formal analysis:** Rabah Debbar, Hamid Boulares.

**Methodology:** Rabah Debbar.

**Project administration:** Abdelkader Moumen.

**Supervision:** Abdelkader Moumen.

**Writing – original draft:** Rabah Debbar, Hamid Boulares, Abdelkader Moumen, Tariq Alraqad, Hicham Saber.

**Writing – review & editing:** Rabah Debbar, Hamid Boulares, Abdelkader Moumen, Tariq Alraqad, Hicham Saber.

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
