## [Decision Letter · Decision Letter 0]

2 Apr 2024

PONE-D-24-09244EXISTENCE AND UNIQUENESS OF NEUTRAL FUNCTIONAL DIFFERENTIAL EQUATIONS WITH SEQUENTIAL FRACTIONAL OPERATORSPLOS ONE

Dear Dr. Moumen,

Thank you for submitting your manuscript to PLOS ONE. After careful consideration, we feel that it has merit but does not fully meet PLOS ONE’s publication criteria as it currently stands. Therefore, we invite you to submit a revised version of the manuscript that addresses the points raised during the review process.

We look forward to receiving your revised manuscript.

Kind regards,

Muhammad Nadeem

Academic Editor

PLOS ONE

Journal Requirements:

5. Please amend your list of authors on the manuscript to ensure that each author is linked to an affiliation. Authors’ affiliations should reflect the institution where the work was done (if authors moved subsequently, you can also list the new affiliation stating “current affiliation:….” as necessary).

Reviewers' comments:

Reviewer's Responses to Questions

**Comments to the Author**

1. Is the manuscript technically sound, and do the data support the conclusions?

Reviewer #1: Yes

Reviewer #2: Yes

2. Has the statistical analysis been performed appropriately and rigorously? 

Reviewer #1: Yes

Reviewer #2: Yes

3. Have the authors made all data underlying the findings in their manuscript fully available?

Reviewer #1: Yes

Reviewer #2: No

4. Is the manuscript presented in an intelligible fashion and written in standard English?

Reviewer #1: Yes

Reviewer #2: Yes

5. Review Comments to the Author

Reviewer #1: Authors considered a hybrid problem under FDEs with generalized form. The work is interesting applicable. I recommend for publication. Some revision should be done like:

1. Why generalized derivative considered? clarify it in introduction in last one paragraphs.

2. Update the literature with recent work like:On theoretical and numerical analysis of fractal--fractional non-linear hybrid differential equations." Nonlinear Engineering 13.1 (2024): 20220372.Using a prior estimate method to investigate sequential hybrid fractional differential equations." Fractals 28.08 (2020): 2040004.Application of topological degree method for solutions of coupled systems of multipoints boundary value problems of fractional order hybrid differential equations." Complexity 2017 (2017).Indian Journal of Pure and Applied Mathematics 51 (2020): 669-687.Indian Journal of Pure and Applied Mathematics 52.1 (2021): 27-38

3. Put some physical applications of hybrid problems.

Reviewer #2: Review Report

Title: EXISTENCE AND UNIQUENESS OF NEUTRAL FUNCTIONAL

DIFFERENTIAL EQUATIONS WITH SEQUENTIAL FRACTIONAL

OPERATORS

Some corrections are needed to improve the quality of the manuscript:

1. What are the prospects of applying the G-Caputo operator derivatives fractional derivatives in practical applications?

2. Abstract should be re write.

3. Clear explanations of classical solutions are expected.

4. Are the theoretical results in the paper dependent on the order of the fractional derivative? If they are independent of the order, then what is the significance of discussing such fractional derivative models?

5. Compared to integer-order models or Riemann-Liouville/Caputo fractional-order models?

6. Authors can give an application for the problem. The authors are expected to provide numerical simulation or numerical example for the considered problem.

7. No proper proof is provided in the example for proving all the Hypotheses are verified. How to verify those results?

8. To show the efficiency of the method, the comparison result with the existing method can be provided.

9. One more example is expected to validate the theory part.

10. Update the bibliography with recent work relevant to the problems in the fractional calculus sense. For example, add the following paper that is closely related to this paper:

a) https://doi.org/10.1016/j.aej.2022.03.011

b) https://doi.org/10.1002/mma.9748

c) https://doi.org/10.1371/journal.pone.0297478

d) doi: 10.3934/math.2024403

e) https://doi.org/10.1016/j.rico.2024.100376

f) DOI:10.5890/DNC.2024.03.014

g) https://doi.org/10.1002/num.22652

Briefly, the authors have derived some interesting results about the governing model. Let the authors do a line-by-line check for correcting the article based on the above suggestions before publication. Therefore, the article can be considered for publication after the major revision.

6. PLOS authors have the option to publish the peer review history of their article (what does this mean?). If published, this will include your full peer review and any attached files.

Reviewer #1: No

Reviewer #2: No

---

## [Author Response · Author response to Decision Letter 0]

1 May 2024

We would like to thank the reviewers for the careful review, and constructive comments and

suggestions. Please see the colored text in the updated version of the manuscript for

incorporated comments and suggestions.

Please find our response in the file labeled 'Response to Reviewers' in section additional files

---

## [Decision Letter · Decision Letter 1]

15 May 2024

EXISTENCE AND UNIQUENESS OF NEUTRAL FUNCTIONAL DIFFERENTIAL EQUATIONS WITH SEQUENTIAL FRACTIONAL OPERATORS

PONE-D-24-09244R1

Dear Dr. Moumen,

We’re pleased to inform you that your manuscript has been judged scientifically suitable for publication and will be formally accepted for publication once it meets all outstanding technical requirements.

Kind regards,

Muhammad Nadeem

Academic Editor

PLOS ONE

Additional Editor Comments (optional):

Reviewers' comments:

Reviewer's Responses to Questions

**Comments to the Author**

1. If the authors have adequately addressed your comments raised in a previous round of review and you feel that this manuscript is now acceptable for publication, you may indicate that here to bypass the “Comments to the Author” section, enter your conflict of interest statement in the “Confidential to Editor” section, and submit your "Accept" recommendation.

Reviewer #1: All comments have been addressed

Reviewer #2: All comments have been addressed

2. Is the manuscript technically sound, and do the data support the conclusions?

Reviewer #1: Yes

Reviewer #2: Yes

3. Has the statistical analysis been performed appropriately and rigorously? 

Reviewer #1: Yes

Reviewer #2: Yes

4. Have the authors made all data underlying the findings in their manuscript fully available?

Reviewer #1: Yes

Reviewer #2: Yes

5. Is the manuscript presented in an intelligible fashion and written in standard English?

Reviewer #1: Yes

Reviewer #2: Yes

6. Review Comments to the Author

Reviewer #1: Accepted as all comments were incorporated. Accepted as all comments were incorporated. .Accepted as all comments were incorporated.

Reviewer #2: Dear Respected Editor

The authors addressed all the comments So I strongly accept and recommend for Publication in your esteemed Journal

Thank you

7. PLOS authors have the option to publish the peer review history of their article (what does this mean?). If published, this will include your full peer review and any attached files.

Reviewer #1: No

Reviewer #2: No

---

## [Editor Report · Acceptance letter]

25 May 2024

PONE-D-24-09244R1 

PLOS ONE

Dear Dr. Moumen, 

I'm pleased to inform you that your manuscript has been deemed suitable for publication in PLOS ONE. Congratulations! Your manuscript is now being handed over to our production team.

Kind regards, 

on behalf of

Dr. Muhammad Nadeem 

Academic Editor

PLOS ONE